# Driving Factors and Scale Effects of Residents' Willingness to Pay for Environmental Protection under the Impact of COVID-19

**Hongkun Zhao** [1,†] **, Yaofeng Yang** [2,†] **, Yajuan Chen** [1,2,*] **, Huyang Yu** [1] **, Zhuo Chen** [1] **and Zhenwei Yang** [3]

1   School of Economics and Management, Inner Mongolia Normal University, Hohhot 010022, China
2   Northwest Institute of Historical Environment and Socio-Economic Development, Shaanxi Normal University, Xi'an 710119, China
3   College of Computer Science and Technology, Inner Mongolia University for Nationalities, Tongliao 028000, China
*   Correspondence: yaya576@126.com
†   These authors contributed equally to this work.

**Abstract:** In recent years, environmental degradation and the COVID-19 pandemic have seriously affected economic development and social stability. Addressing the impact of major public health events on residents' willingness to pay for environmental protection (WTPEP) and analyzing the drivers are necessary for improving human well-being and environmental sustainability. We designed a questionnaire to analyze the change in residents' WTPEP before and during COVID-19 and an established ordinary least squares (OLS), spatial lag model (SLM), spatial error model (SEM), geographically weighted regression (GWR), and multiscale GWR to explore driver factors and scale effects of WTPEP based on the theory of environment Kuznets curve (EKC). The results show that (1) WTPEP is 0–20,000 yuan before COVID-19 and 0–50,000 yuan during COVID-19. Residents' WTPEP improved during COVID-19, which indicates that residents' demand for an ecological environment is increasing; (2) The shapes and inflection points of the relationships between income and WTPEP are spatially heterogeneous before and during COVID-19, but the northern WTPEP is larger than southern, which indicates that there is a spatial imbalance in WTPEP; (3) Environmental degradation, health, environmental quality, and education are WTPEP's significant macro-drivers, whereas income, age, and gender are significant micro-drivers. Those factors can help policymakers better understand which factors are more suitable for macro or micro environmental policy-making and what targeted measures could be taken to solve the contradiction between the growing ecological environment demand of residents and the spatial imbalance of WTPEP in the future.

**Keywords:** COVID-19 pandemic; willingness to pay for environmental protection; environmental Kuznets curve; multiscale geographically weighted regression; spatial econometrics; spatial heterogeneity

## 1. Introduction

Since the United Nations adopted the 2030 Agenda for Sustainable Development in 2015, sustainable development has become the goal of all countries around the world, including extreme poverty and hunger eradication, improvement of health and well-being, and environmental protection [1]. However, compared to the pre-COVID-19 era, mankind is facing many serious challenges in terms of health, resources, environment and economy [2–5], which has raised great concerns about the harmonious coexistence between humans and nature [6]. Therefore, research on the subject of environmental protection has become a focus of current research in sustainable development and ecological environment, and has gained attention from governments around the world [7]. In this context, modeling residents' willingness to pay for environmental protection (WTPEP) is of significant meaning for the implementation of government environmental policies and instruments.

Willingness to pay refers to the maximum monetary amount that a unit is willing to pay for acquiring merchandise [8]. With increasing environmental challenges in recent years, willingness to pay has also been extended to describe the WTPEP in the field of ecological environment and sustainable development [9,10], which has tended to address two main objectives. One is market research: measuring willingness to pay for an environmental-related good or service across a population is tantamount to estimating its market demand [11,12]. The other is to support policy analysis: whether published in the literature on sustainable development, compensation for ecological conservation, or environmental economics, a key motivation for many of these studies has been to support cost-benefit evaluations of private or public environmental interventions [13,14]. However, mounting evidence shows that identifying key drivers of WTPEP is an interesting, complex, and controversial process. There is a direct link between income and WTPEP, as poverty and hunger will affect the upper limit value to pay for environmental protection [15]. In related studies, the type of income that impacts the residents' WTPEP can be divided into two categories: one is macro income level or regional per capita disposable income, and the other is micro income level or residents' individual income level [16]. Among the research on this topic, most existing studies model the relationship between WTPEP and income based on an Environment Kuznets Curve (EKC) hypothesis.

Kuznets proposed the Kuznets curve hypothesis in 1995, which refers to the inverted U-shaped relationship between per capita income and income inequality [17]. As income per capita increases at the initial stages, income inequality follows the same path but begins to decline after reaching a turning point. Originating from the simulation of Kuznets curve, the EKC hypothesis was proposed by Grossman and Krueger to examine the impact of economic development on the degree of environmental pollution [18]. Grossman and Krueger found an inversed-U-shaped relationship between air pollutants and income, thus, confirming the effectiveness of the EKC hypothesis created by Panayotou [19]. WTPEP is an environmental behavior of residents closely related to environmental pollution. Thus, more existing studies suggest that the relationship between WTPEP and income is characterized by EKC, which means that as per capita income increases, WTPEP decreases at the initial stage of economic development and will increase after a certain turning point [20]. However, the results of EKC studies are complex and controversial. Researchers not only found inconsistent numerical values for the position of the inflection points but also found inconsistencies in the shapes and other characteristics of curves [21–23]. One likely explanation for the above contradictory results is that the behavior of a given curve characteristic is influenced by other driving factors [24]. In terms of demographic characteristics, educational level, gender, and age explain a biased result which suggests strong evidence for an upward assumptive deviation in evaluating the WTPEP for road freight noise reduction [25]. At the aspect of environmental conditions, the translation of environmental quality attitude into WTPEP was less when the same level of environmental quality attitude was weak rather than strong [26]. In terms of health, the WTPEP of residents who are more affected by environmental pollution is greater than those whose health is less affected by environmental pollution [27]. The above shows that income, demographic characteristics, local environmental conditions, and health are the key driving factors of residents' WTPEP.

Many studies have investigated the driving factors of WTPEP using different methods, which is significant for clarifying the mechanism of WTPEP. According to relevant studies, these methods can be divided into non-spatial models and spatial models. Non-spatial models mainly consist of ordinary least squares (OLS), logit or probit model [14], Tobit regression model [28], stepwise regression model [29], propensity score matching [30], quantile regression [31], and regression discontinuity design [7]. In contrast, spatial models mainly consist of spatial error model (SEM) [32], spatial lag model (SLM) [33], spatial Dubin model [34], and geographically weighted regression (GWR) [35]. The non-spatial regression models assume that the coefficients of different regions are unchanged, and only reflect the average influence of each factor, ignoring its potential spatial effect. In reality, the change in WTPEP is characterized by an interleaved chaotic distribution of

high and low values [36]. Because the geographical scope of environmental problems is large, individual environmental protection behavior alone cannot bring obvious local benefits [37]. Environmental protection requires the joint efforts of every member, and only when there are enough individuals participating can it have a noticeable effect. Modeling spatial regression equations for each individual to consider the spatial relationship of driving factors, which can more accurately describe WTPEP than the non-spatial regression model [38]. However, it is still necessary to further study the scale effects of WTPEP drivers to enrich the theoretical research on WTPEP mechanisms and formulate scientific environmental policies, which is a novel research attempt.

Another innovation of this study is investigating the impact of major public health events on residents' WTPEP and its mechanism. The COVID-19 pandemic changed the influencing factors of residents' WTPEP in regard to human behavior, personality traits, mental health, socioeconomic status, and the environment [2,7]. Thus, it is necessary to study the change in residents' WTPEP before and during COVID-19, which will be significant in advising governments' plans for a post-pandemic recovery worldwide and responding to major public health events [39]. To provide more insights into these phenomena, we designed a study to answer three main questions: (1) revealing the change in Chinese WTPEP before and during COVID-19; (2) clarifying whether the EKC relationship exists in WTPEP; (3) exploring the driving factors and scale effects of WTPEP based on the theory of EKC. Based on this study, we try to enrich the theory of the driving mechanism of residents' WTPEP and provide new academic references for improving residents' WTPEP and dealing with the impact of major public health events on WTPEP.

## 2. Regional Overview and Dataset

### 2.1. Regional Overview

This survey covers 31 provinces in China. China lies in the east of Asia and west of the Pacific Ocean. Figure 1 depicts the location of the study area and the distribution of officially published confirmed COVID-19 cases by province up to 31 August 2022 (the cut-off date for this study survey). Affected by environmental factors, socioeconomic activities, and medical conditions, the number of confirmed COVID-19 cases in eastern China is far greater than in western China [40]. In particular, Inner Mongolia, Shaanxi, Hubei, and Guangdong have the highest number of COVID-19 cases in China and are regions with the highest prevalence of COVID-19. The number of confirmed cases in each province has a differential impact on local residents' WTPEP due to varying degrees of the threat posed by the pandemic to local economic development and residents' health. Therefore, it is important to analyze the drivers and scale effects of residents' WTPEP before and during COVID-19 based on the extent of the pandemic and the spatial perspective of provinces, which will help provincial governments formulate effective local environmental policies [41].

### 2.2. Questionnaire Design

To understand WTPEP, we designed a questionnaire to collect data (Table 1). The questionnaire included eleven questions with two being pre-questions used to quantify WTPEP (Q1–2). The remaining nine questions were targeted at understanding the influencing factors of WTPEP and were divided into five categories. The first part (one question) aimed to explore the spatial relationship between WTPEP and its influencing factors by gathering respondents' geographic positions (Q3). Based on the EKC theory, the second part (one question) was set to explore the relationship between WTPEP and income by gathering respondents' income (Q4). The third part (three questions) was set to gather respondents' demographic characteristics (Q5–Q7). The fourth part (two questions) was set to gather the environmental conditions of respondents' location (Q8–Q9). The fifth part (two questions) was set to gather the level of environmental degeneration impacts on respondents' health before and during COVID-19 (Q10–Q11).

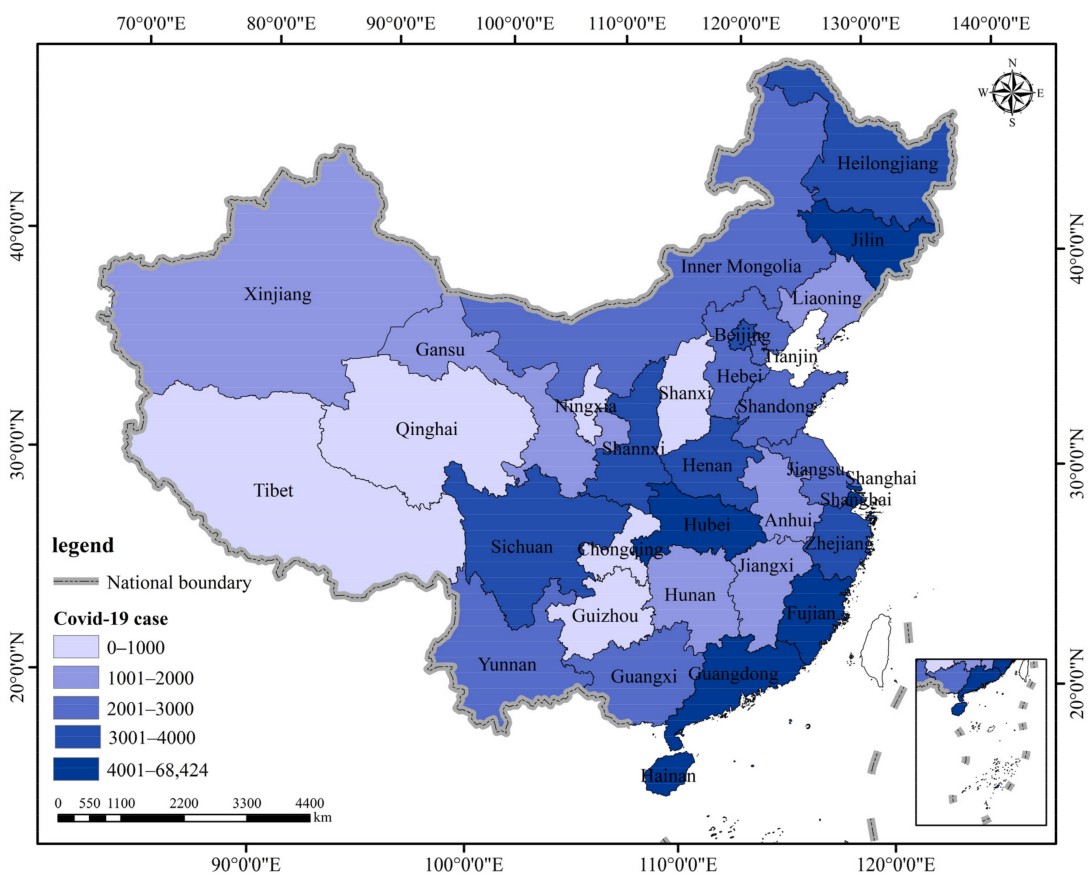

**Figure 1.** Location of the study area and distribution of confirmed cases of COVID-19 up to 31 August 2022.

**Table 1.** Content of the questionnaire survey.

| Question | Result (Only One Response per Question Is Permitted) |
|---|---|
| **WTPEP (yuan)** | |
| How much money would you be willing to pay for environmental protection before COVID-19? | Quantity (yuan) |
| How much money would you be willing to pay for environmental protection during COVID-19? | Quantity (yuan) |
| **Location** | (longitude, dimension) |
| **Net annual income (ten thousand yuan)** | ≥0 |
| **Demographic characteristics** | |
| Age | >0 |
| Gender | Man, Woman |
| Educational level | primary school (6 years), middle school (9 years), high school (12 years), college or university (16 years), master or above (19 years) |
| **Environmental condition** | |
| What do you think of the quality of the environment at your location? | very good (5), good (4), general (3), bad (2), very bad (1) |
| What do you think of the degree of environmental degradation at your location? | extremely serious (5), very serious (4), serious (3), not serious (2), don't know (1) |
| **Affect Health** | |
| How strongly does environmental degradation affect your health before COVID-19? | extremely serious (5), very serious (4), serious (3), not serious (2), don't know (1) |
| How strongly does environmental degradation affect your health during COVID-19? | extremely serious (5), very serious (4), serious (3), not serious (2), don't know (1) |

*2.3. Data Collection*

The survey was conducted three times using a staged sampling method during the COVID-19 pandemic, which was from May to June 2020, from April to May 2021, and

August 2022, respectively. Before the large-scale official survey, we conducted a preliminary survey in Inner Mongolia and Shanxi. Based on the respondents' feedback and comparisons between survey data and official data, we confirmed the scientific content of the questionnaire and the feasibility of the survey method. Next, we used the 31 provinces as sampling units to conduct a spatial quota sampling based on the population size of each province and then conducted a web-based survey using QR codes (an electronic survey tool generated based on a questionnaire survey platform) to collect responses from each province. To avoid human error and duplicate responses, we set completion limits and allowed each IP address only one opportunity to complete the questionnaire. This method has four major advantages: rapid dissemination, no temporal or spatial limitations, quality monitoring, and high objectivity. In total, we collected 1018 questionnaires and screened 1009 high-quality questionnaire datasets which covered all 31 provinces.

When the number of potential respondents is sufficiently large, the minimum sample size available for the study is only affected by error and confidence levels, because there is no necessary link between the sample size and the total population [42]. The minimum sample size can be calculated using Formula (1):

$$n = \frac{T^2 \alpha^2}{\beta^2} \tag{1}$$

where $n$ indicates the sample size; $T$ denotes the statistics under a certain level of confidence. $\alpha$ is the standard deviation of the population and is usually set to 0.5. $\beta$ is the allowable error. In general, a confidence level of 90% and an allowable error of 3% are appropriate for the sample. We acknowledge that the minimum sample size required is 752 [43], which confirms that our sample size is sufficient.

The ratios of a sample size to the provincial population are shown in Supplementary Section S1 and S2 (Figures S1 and S2). Except for Inner Mongolia and Shanxi which are included pre-researched, only Tibet and Qinghai had higher ratios compared to other provinces. This is mainly due to their large land area but low total population and population density. To ensure the spatial uniformity of sample distribution, we increased the sample size of Tibet and Qinghai accordingly. The data from the questionnaire survey were compared with the official data published by the National Bureau of Statistics of China, as shown in Table 2. By comparing the sample data, official data, and the actual societal situation, we confirmed that the survey is representative and scientific in Supplementary Section S3.

**Table 2.** Descriptive statistics of variables.

| Variable | Obs | Mean | St.Dev. | Min | Max | Official Data |
|---|---|---|---|---|---|---|
| **Dependent variable** | | | | | | |
| WTPEP (before COVID-19) (yuan) | 1009 | 1224.294 | 1666.231 | 0 | 20,000 | 1843 |
| WTPEP (during COVID-19) (yuan) | 1009 | 1967.389 | 2648.569 | 0 | 50,000 | 2120 |
| **Independent variable** | | | | | | |
| Age | 1009 | 31.304 | 11.896 | 12 | 86 | 38.8 |
| Gender | 1009 | 0.409 | 0.492 | 0 | 1 | 0.512 |
| income (yuan) | 1009 | 41,375.26 | 53,188.218 | 0 | 40 | 36,883 |
| Edu | 1009 | 15.186 | 3.242 | 6 | 19 | 9.91 |
| EQ | 1009 | 3.51 | 0.838 | 1 | 5 | |
| ED | 1009 | 2.573 | 0.835 | 1 | 5 | |
| health (before COVID-19) | 1009 | 2.535 | 1.01 | 1 | 5 | |
| health (during COVID-19) | 1009 | 2.77 | 1.106 | 1 | 5 | |

## 3. Model Specification and Methods

### 3.1. Theoretical Model Setting

EKC describes an inverted U-shaped relationship between economic development and environmental degradation, where environmental degradation refers to the destructive

effects of human actions on the environment [17]. WTPEP is an environmental behavior of residents that is closely related to environmental degradation. Therefore, we innovatively used EKC to study the relationship between WTPEP and income. The EKC relationships between income and WTPEP are shown in Figure 2. As shown in the I part of Figure 2: individuals whose income is on the left side of the inflection point of the U-shaped curve mainly have insufficient income to fully support production and life, so the role of environmental protection willingness is not particularly obvious, and the WTPEP of residents on the left side of the inflection point shows a downward trend. When income increases to the right side of the inflection point, environmentally conscious residents are willing to invest surplus money in environmental protection, so the WTPEP of residents on the right half of the curve shows an upward trend with income. As shown in part II of Figure 2: if residents' environmental awareness is not strong, they will invest more money to expand production and improve material consumption resulting in a declining trend of WTPEP for residents on the right half of the curve [23]. As shown in part III of Figure 2: individuals whose income is on the left side of the inflection point of the inverted U-shaped curve mainly have better environmental awareness and are willing to increase their expenditure on environmental protection while obtaining more income. The WTPEP of residents on the right side of the inflection point shows a downward trend, and the increase in individual income expands the income inequality between individuals. High-income people with poor environmental awareness will increase production input and expand the income gap, causing the elasticity of income to WTPEP to continue to decline even if the elasticity is less than zero. As shown in part IV of Figure 2: if residents' environmental awareness is particularly strong, they will invest more money in environmental protection, resulting in a continued upward trend of WTPEP for residents on the right half of the curve [20].

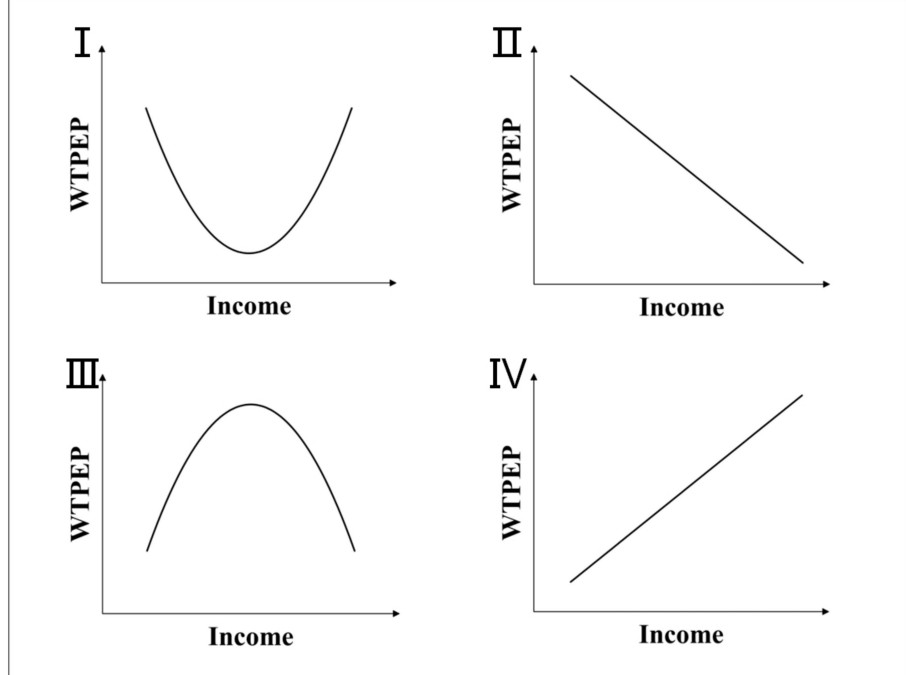

**Figure 2.** Various EKC characteristics between WTPEP and income [37].

In Table 2, we identify some factors that influence residents' WTPEP based on the Chinese COVID-19 background and the theory of EKC [23,44–47]. We used multiple linear regression model to evaluate the significance *p*-value and factors' VIF index, and further eliminate the factors that have high redundancy. The model is shown in Equation (2):

$$WTPEP_i = \beta_0 + \beta_1 income_i + \beta_2 (income_i)^2 + \beta_3 gender_i + \beta_4 age_i + \beta_5 Edu_i + \beta_6 EQ_i + \beta_7 ED_i + \beta_8 health_i + \varepsilon_i \quad (2)$$

where WTPEP indicates residents' WTPEP before or during COVID-19; income stands for residents' net annual income; $(income)^2$ represents the square of residents' net annual income; gender is resident's gender; age donates resident's age; education stands for resident's educational level; EQ is environmental quality at resident's location; ED represents the level of environmental degradation at resident's location; health stands for the degree of environmental degradation which affects resident's health before or during COVID-19; i represent the i-respondent; $\beta_0$ is the constant term and $\beta_1$–$\beta_8$ are parameters to be estimated; $\varepsilon_i$ is the random error term. Equation (2) enables tests for various relationships between income and WTPEP as follows:

I.    U-shaped relationship between income and WTPEP if $\beta_1 < 0$, $\beta_2 > 0$;
II.   A negative monotonic relationship between income and WTPEP if $\beta_1 < 0$, $\beta_2 = 0$;
III.  Inversed-U shaped relationship between income and WTPEP if $\beta_1 > 0$, $\beta_2 < 0$;
IV.   A positive monotonic relationship between income and WTPEP if $\beta_1 > 0$, $\beta_2 = 0$.

*3.2. Global Spatial Regression Modeling*

Although assuming that the WTPEP is independent of each other, the linear regression model omits the possibility of spatial correlation among many explanatory variables in the model [48]. Therefore, two spatial autoregressive models (SLM and SEM) are based on linear regression and take into account spatial weights and dependencies.

3.2.1. Spatial Lag Model (SLM)

The SLM model assumes that there are dependencies between dependent and independent variables, and can be used to estimate spatial spillover effects [49]. The theoretical model of SLM is as follows:

$$y_i = \beta_0 + \beta_1 x_i + \rho w_i y_i + \varepsilon_i \tag{3}$$

where $\rho$ is the spatial lag coefficient, and w is a spatial weights matrix that demonstrates the distance relationship between observations i and j. The $\rho W_i$ depicts the spillover effects from adjacent units of individual i.

3.2.2. Spatial Error Model (SEM)

Compared to the SLM model, the SEM indicates spatial correlation through the distribution of errors over spatial units [50]. The SEM model is expressed as follows:

$$y_i = \beta_0 + \beta_1 x_i + \lambda(W_i)\mu_i + \varepsilon_i \tag{4}$$

where at resident i, $\mu_i$ indicates error's spatial component, $\lambda$ indicates the degree of relatedness between components, and $\varepsilon_i$ is a spatial uncorrelated error term.

*3.3. Local Spatial Regression Modeling*

3.3.1. Geographically Weighted Regression (GWR)

Compared to global models, GWR allows parameters to vary spatially and assumes that non-stationary relationships exist between the response variable and explanatory variables. The GWR model is calculated as follows [51,52]:

$$y_i = \beta_{0i}(u_i, v_i) + \sum_{n=1}^{k} \beta_{ni}(u_i, v_i)x_{ni} + \varepsilon_i \tag{5}$$

where, $y_i$ denotes the i-resident's WTPEP; xni represents the i-resident's nth factor; $\beta_{0i}$ is the intercept; $\beta_{ni}$ is the regression coefficient; $(u_i, v_i)$ indicates i-resident's geographical coordinates. Estimates of parameters for each independent variable and each WTPEP in matrix form is given by:

$$\hat{\beta}(i) = \left(X'W(i)X\right)^{-1}X'W(i)y \tag{6}$$

where $\hat{\beta}$ indicates parameter estimation vector (p × 1), X displays the matrix of the selected independent variable (n × p), $W_{(i)}$ is the spatial weight matrix (n × n), and y implies WTPEP's vector observation (p × 1).

### 3.3.2. Multiscale Geographically Weighted Regression (MGWR)

In many cases, the different spatial influences of residents' WTPEP are involved with varying spatial scales. Compared to GWR, MGWR allows the relationship between independent variables and dependent variable to vary spatially at different scales, and is calculated as follows [53]:

$$y_i = \sum_{j=0}^{m} \beta_{bwj} x_{ij} + \varepsilon_i \tag{7}$$

where $\beta_{bwj}$ represents bandwidths, which are utilized to calibrate the *j*th conditional relationship. Compared to GWR, the model has several advantages, such as it can accurately depict spatial heterogeneity, diminish collinearity, and lessen the bias in the parameter estimates [54].

### 3.4. Model Fitting

A range of income, demographic, environmental, and health variables were included in the modeling process to determine which factors are associated with WTPEP. VIF was used to test the multicollinearity among variables, and independent variables without correlation were selected as model variables [55]. A spatial weight matrix was generated based on first-order Queens' contiguity to reflect how individuals affect each other spatially. By developing the local models, the kernel function type of adaptive bi-square with its bandwidth size was specified. The corrected Akaike Information Criterion (AICc) was used to select the optimal bandwidth. Global and local models operate in GeoDa 1.20 and MGWR 2.2, respectively. The best model fit is indicated by a larger adjusted R2 and a smaller AICc value [56,57].

## 4. Results

### 4.1. Changes in WTPEP before and during COVID-19

Before and during COVID-19, Chinese WTPEP changed considerably (Figure 3). Before COVID-19, WTPEP's average was 1224.29 yuan, maximum value was 20,000 yuan (Table 2). During COVID-19, WTPEP's average turned to 1967.39 yuan, year-on-year growth of 60.7%, and maximum value was 50,000 yuan, year-on-year growth of 150.0%. The minimum value was 0 yuan both before and during COVID-19. The spatial distribution of WTPEP was relatively scattered and did not have strong regularity. In general, the northern WTPEP was higher than the southern. Regarding the spatial characteristics of WTPEP, Shanghai had the largest mean of 3438.89 yuan, whereas Hubei had the smallest mean of 635 yuan. During COVID-19, WTPEP mostly increased except in Xinjiang, Xizang, Qinghai, Yunnan, and Guizhou. The province with the largest increase in average was Zhejiang, which increased from 3300 to 5500 yuan. The province with the least increase in average was Qinghai, which decreased from 2968.75 yuan to 1393.75 yuan. In general, WTPEP showed an upward trend with high and low values scattered in the spatial distribution complicatedly during COVID-19.

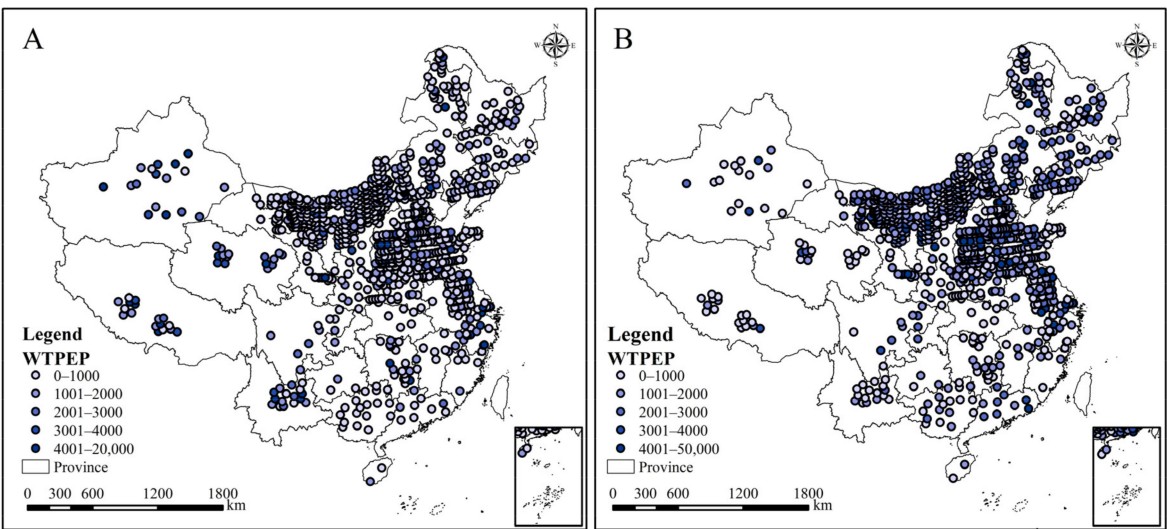

**Figure 3.** Spatial distribution of Chinese WTPEP before and during COVID-19. (**A**) stands for before COVID-19, (**B**) represents during COVID-19.

### 4.2. Model Choice

The outcomes of linear regression model are shown in Table 3. After removing variable (income)³, VIF values of all independent variables showed low multicollinearity (all less than threshold 5), which indicates that there was no multicollinearity among variables. Before COVID-19, the coefficient estimates of income, (income)², Edu, ED, and health were statistically significant with *p*-values less than 0.01. During COVID-19, the coefficient estimates of (income)², Edu and EQ were significant with *p*-values less than 0.01, indicating a mix of positive and negative associations. For instance, the coefficient estimates of EQ were negative, suggesting that a decrease in environmental quality is associated with an increased WTPEP. The coefficient estimates also indicate that ED was the most influential variable, followed by income and Edu before COVID-19. During COVID-19, EQ was the most influential variable, followed by Edu and (income)². The linear regression model shows a low adjusted R² (0.39; 0.25), indicating that 61% or 75% of the variance of WTPEP is caused by unknown variables (Table 4). Some driving factors present statistically nosignificance, and some results are inconclusive, so further research should take into account the spatial nonstationarity between response variables and explanatory variables based on spatial models.

**Table 3.** Summary statistics of the linear regression model.

| Variable | Coefficient | | St. Error | | Probability | | VIF | |
|---|---|---|---|---|---|---|---|---|
| | **Before** | **During** | **Before** | **During** | **Before** | **During** | **Before** | **During** |
| Intercept | −1061.15 | 966.23 | 433.72 | 760.76 | 0.015 | 0.204 | —— | —— |
| income | 66.74 | 0.97 | 17.07 | 30.00 | 0.000 | 0.974 | 4.88 | 4.90 |
| (income)² | 5.64 | 11.01 | 0.76 | 1.34 | 0.000 | 0.000 | 4.69 | 4.73 |
| Age | 6.34 | 8.23 | 4.27 | 7.47 | 0.138 | 0.271 | 1.53 | 1.52 |
| Gender | 135.92 | 86.23 | 84.60 | 148.22 | 0.108 | 0.561 | 1.03 | 1.02 |
| Edu | 49.66 | 63.03 | 15.34 | 26.88 | 0.001 | 0.019 | 1.46 | 1.46 |
| EQ | 51.25 | −204.24 | 52.24 | −0.07 | 0.327 | 0.026 | 1.13 | 1.13 |
| ED | 145.30 | −107.50 | 56.60 | −0.03 | 0.010 | 0.262 | 1.32 | 1.23 |
| health | 75.52 | 86.98 | 45.02 | 0.04 | 0.094 | 0.209 | 1.23 | 1.12 |

**Table 4.** Comparison of the goodness of fit measures for global and local models.

| Criterion | OLS | | SLM | | SEM | | GWR | | MGWR | |
|---|---|---|---|---|---|---|---|---|---|---|
| | Before | During | Before | During | Before | During | Before | During | Before | During |
| Adj.$R^2$ | 0.39 | 0.25 | 0.40 | 0.27 | 0.40 | 0.27 | 0.62 | 0.56 | 0.68 | 0.63 |
| AICc | 17,348.7 | 18,482.1 | 17,337.2 | 18,480.0 | 17,344.6 | 18,474.9 | 2161.9 | 2180.2 | 1863.3 | 2045.5 |

Moran's I was used to verifying if there is a spatial autocorrelation between WTPEP before and during COVID-19. As shown in Table 5, the z-value for Moran's I test is positive and significant, which indicates the presence of positive spatial autocorrelation in WTPEP. As a result, the basic linear regression model was also re-estimated correcting for spatial autocorrelation.

**Table 5.** Moran's Tests for WTPEP before and during COVID-19.

| Period | Moran's *I* Statistic | Variance | *Z*-Value | *p*-Value |
|---|---|---|---|---|
| Before COVID-19 | 0.128 | 0.000 | 7.054 | 0.000 |
| During COVID-19 | 0.043 | 0.000 | 2.565 | 0.005 |

SLM and SEM were used to characterize the relationship between factors and WTPEP, which improved the accuracy of the global modeling. Before COVID-19, the lag coefficients were strongly positive ($p < 0.01$) and the adjusted $R^2$ of SLM and SEM increased from 0.39 to 0.40 and 0.40, respectively (Table 4), whereas the AICc slightly decreased. During COVID-19, the lag coefficients were strongly positive ($p < 0.05$) and the adjusted $R^2$ of SLM and SEM increased from 0.25 to 0.27 and 0.27, respectively (Tables 4 and 6), whereas the AICc slightly decreased. Although SLM and SEM provide a closer fit than OLS according to the adjusted $R^2$, the fitting results are still not optimistic, which can be attributed to the neglected scale of spatial processes involved in modeling WTPEP. Accordingly, a spatially non-stationary local modeling approach was adopted next.

To explore the local spatial variation in the relationships with residents' WTPEP, GWR and MGWR were applied to the same set of predictors used in global models. The diagnostics of GWR indicated a relatively improved adjusted $R^2$ and AICc (Table 4): before COVID-19, fitting GWR using the optimal bandwidth of 111, the adjusted $R^2$ increased to 0.62, whereas the AICc decreased to 2161.9; during COVID-19, fitting GWR using the optimal bandwidth of 197, the adjusted $R^2$ increased to 0.56, and the AICc decreased to 2180.2. However, among all fitted models before COVID-19, MGWR represents the largest adjusted $R^2$ (0.68) and the lowest AICc (1863.3). Among all fitted models during COVID-19, MGWR represents the largest adjusted $R^2$ (0.63) and the lowest AICc (2045.5). In summary, by comparing the global spatial regression model and local spatial regression model, MGWR is the optimal model to explore WTPEP's driving mechanism (Table 4).

**Table 6.** Summary statistics of SLM, and SEM for WTPEP before and during COVID-19.

| Variable | Coefficient | | | | St. Error | | | | Z-Score | | | |
| --- | --- | --- | --- | --- | --- | --- | --- | --- | --- | --- | --- | --- |
| | Before | | During | | Before | | During | | Before | | During | |
| | SLM | SEM | SLM | SEM | SLM | SEM | SLM | SEM | SLM | SEM | SLM | SEM |
| Intercept | −1137.36 *** | −977.59 ** | 738.65 | 813.20 | 429.02 | 434.33 | 759.70 | 761.93 | −2.65 | −2.25 | 0.97 | 1.07 |
| income | 56.04 *** | 63.18 *** | 1.09 | 13.40 | 16.62 | 16.74 | 29.20 | 29.44 | 3.37 | 3.77 | 0.04 | 0.46 |
| (income)$^2$ | 5.84 *** | 5.64 *** | 10.96 *** | 10.60 *** | 0.74 | 0.75 | 1.31 | 1.31 | 7.87 | 7.55 | 8.35 | 8.07 |
| age | 6.29 | 5.92 | 8.01 | 7.43 | 4.20 | 4.26 | 7.39 | 7.46 | 1.50 | 1.39 | 1.08 | 1.00 |
| gender | 133.51 | 135.72 | 97.68 | 123.50 | 83.47 | 84.04 | 147.15 | 146.88 | 1.60 | 1.61 | 0.66 | 0.84 |
| Edu | 46.83 *** | 47.41 *** | 61.70 ** | 62.27 ** | 15.13 | 15.35 | 26.68 | 26.92 | 3.10 | 3.09 | 2.31 | 2.31 |
| EQ | 54.01 | 52.95 | −194.29 ** | −189.64 ** | 51.56 | 52.07 | 91.04 | 91.21 | 1.05 | 1.02 | −2.13 | −2.08 |
| ED | 135.59 ** | 136.66 ** | −97.18 | −83.39 | 55.86 | 56.71 | 95.09 | 96.09 | 2.43 | 2.41 | −1.02 | −0.87 |
| health | 72.33 | 76.91 ** | 89.25 | 98.88 | 44.44 | 44.98 | 68.62 | 68.87 | 1.63 | 1.71 | 1.30 | 1.43 |
| Rho | 0.15 *** | | 0.09 ** | | 0.04 | | 0.05 | | 3.65 | | 2.02 | |
| Lambda | | 0.11 ** | | 0.14 *** | | 0.05 | | 0.05 | | 2.07 | | 2.73 |

Notes: ** indicates significance at 5% level; *** indicates significance at 1% level.

### 4.3. Spatial Heterogeneity of EKCs' Shapes and Inflection Points

4.3.1. Spatial Heterogeneity Analysis of EKCs' Shapes

Figure 4 reveals spatial heterogeneity in the EKCs' shapes, EKCs' inflection points, and comparison of incomes and inflection points before and during COVID-19. Before COVID-19, the EKCs have four shapes: showing a negative relationship between income and WTPEP in Hebei; indicating an inverted U-shaped relationship between income and WTPEP in southwest and northeast China; representing a U-shaped relationship between income and WTPEP in Hebei, Shandong, Jiangsu and central Inner Mongolia; meaning a positive relationship between income and WTPEP in Northeast of central China. During COVID-19, EKCs have three shapes: standing for an inverted U-shaped relationship between income and WTPEP in southern China; representing a U-shaped relationship between income and WTPEP in northwest China; showing a positive relationship between income and WTPEP in the northeast of central Hebei, Liaoning, and Inner Mongolia.

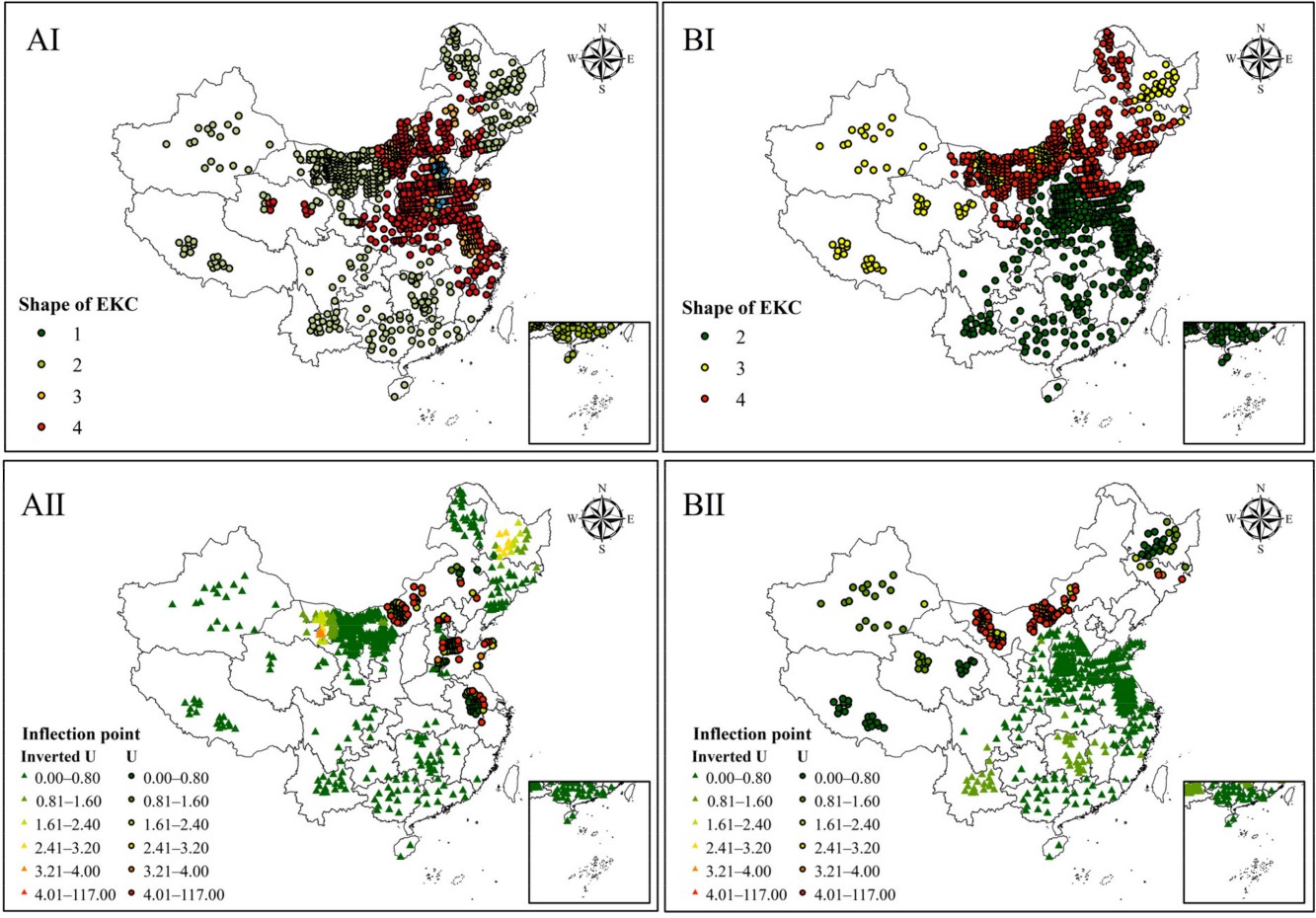

**Figure 4.** *Cont.*

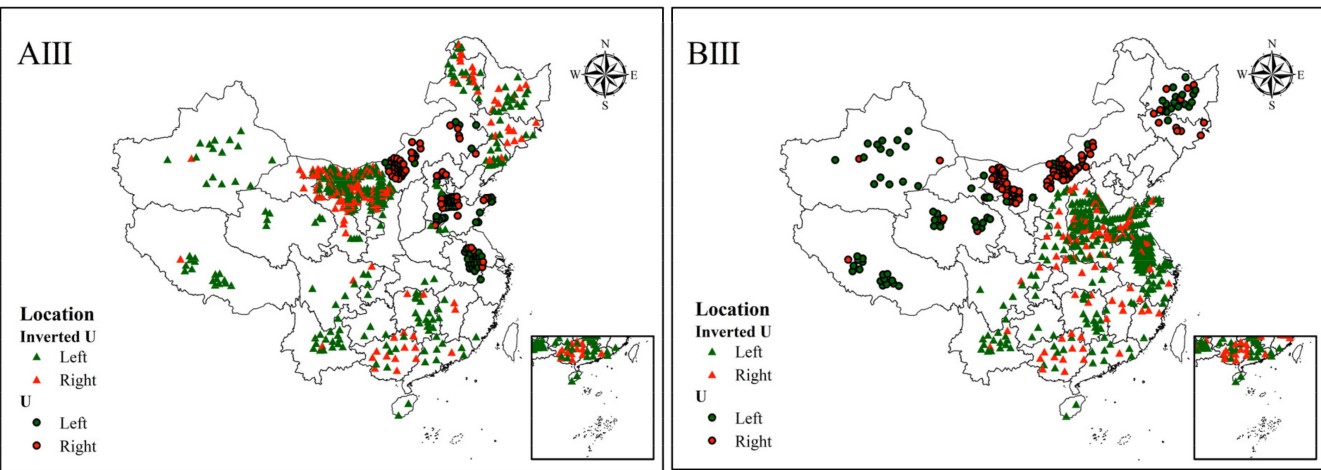

**Figure 4.** The EKCs' shapes (**upper**), inflection points (**middle**), and where the current income positions are relative to the inflection points (**lower**). Notes: In the upper two panels, 1 represents a negative relationship between income and WTPEP; 2 indicates an inverted U-shaped relationship between income and WTPEP; 3 means a U-shaped relationship between income and WTPEP; 4 stands for a positive relationship between income and WTPEP. (**AI–AIII**) stands for before COVID-19, (**BI–BIII**) represents during COVID-19, the same below.

4.3.2. Spatial Heterogeneity Analysis of EKCs' Inflection Points

Before COVID-19, the inflection points of inverted U-shaped EKCs were 1.53 to 36,886.45 yuan, the mean was 5437.32 yuan (Figure 4(AII)), and the income levels of residents in the western region were on the left sides of the inflection points (Figure 4(AIII)). Before COVID-19, the inflection points of U-shaped EKCs were 1160.06 to 1,159,827.67 yuan, and the mean was 48,706.30 yuan. During COVID-19, the inflection points of inverted U-shaped EKCs were 104.79–13,580.05 yuan, the mean was 3647.34 yuan (Figure 4(BII)), and the income levels of the Central Plains were on the right sides of the inflection points (Figure 4(BIII)). Before COVID-19, the inflection points of U-shaped EKCs were 3739.28–1,161,339.79 yuan, the mean was 62,437.36 yuan, and the income levels of residents in the western region were on the left sides of the inflection points.

*4.4. Spatial Heterogeneity and Scale Effect Analysis of WTPEP's Drivers*

4.4.1. Spatial Heterogeneity of WTPEP's Drivers

Before COVID-19, age had a positive impact on residents' WTPEP, indicating that older people had more WTPEP than younger people (Figure 5). But there was a negative relationship between age and WTPEP during COVID-19, showing that young people's awareness of environmental protection had improved. The spatial distribution of age's coefficients increased gradually from east to west before COVID-19 but decreased gradually during COVID-19. The change in gender's coefficient spatial distribution was minimal except in Xizang, Xinjiang, and Yunnan during COVID-19 (Figure 5), indicating that men's WTPEP decreased compared with women in the three provinces. Men's WTPEP decreased from south to north compared with women. Other things being equal, men in south China have more WTPEP than women, but men have less WTPEP than women in northern China.

The educational level has a positive effect on residents' WTPEP except in eastern China before COVID-19 (Figure 6). During COVID-19, the coefficients of educational level in eastern China increased and were higher than those in western China which decreased, and were positive all over China. Environmental quality was positively correlated with residents' WTPEP before COVID-19 and had a greater impact on residents' WTPEP in the midland than in eastern and western China (Figure 6). Local residents with better environmental quality will be more WTPEP, in turn, the greater residents' WTPEP, the better environmental quality will be. However, during COVID-19, there was a positive

correlation between environmental quality and WTPEP in southeast China, but a negative correlation in northwest China.

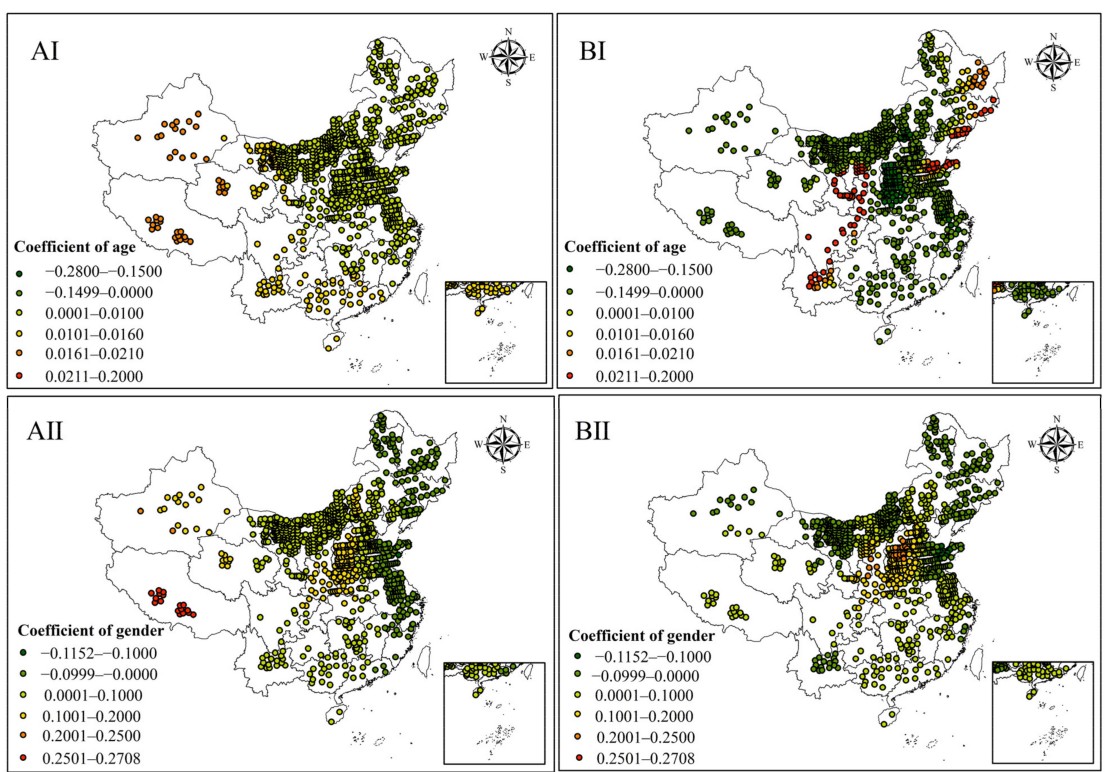

**Figure 5.** The effects of age (**above**) and gender (**below**) in describing WTPEP before (**left**) and during (**right**) COVID-19 based on MGWR models.

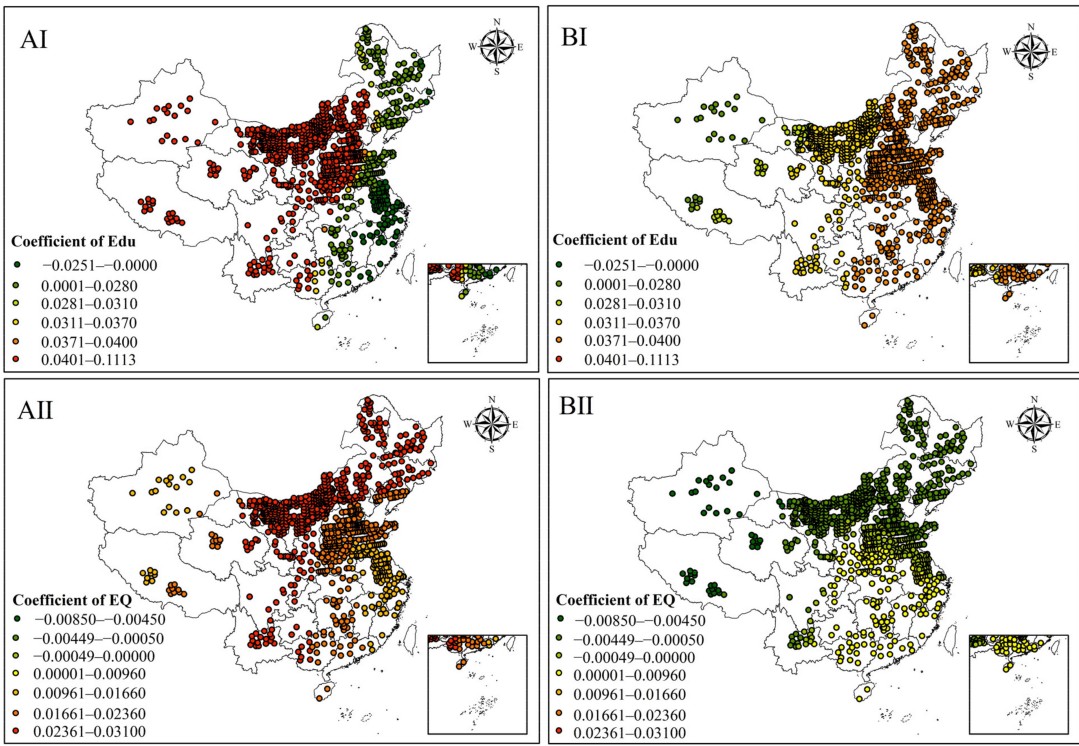

**Figure 6.** The effects of education (**above**) and environmental quality (**below**) in describing WTPEP before (**left**) and during (**right**) COVID-19 based on MGWR models.

Environmental degradation degree is positively correlated with WTPEP, the greater the environmental degradation degree, the stronger residents' WTPEP, and the impact of environmental degradation on WTPEP declined slightly during COVID-19 (Figure 7). The environmental degradation coefficients of the regions with good environmental quality are smaller than those of regions with bad environmental quality. For example, the environmental quality in southern China is better than those in northwest China, but environmental degradation coefficients are smaller than those in northwest China. The degree of environmental degradation impact on health is positively correlated with WTPEP (Figure 7), the greater degree of environmental degradation impact on health, the stronger residents' WTPEP. During COVID-19, the coefficient of health increased slightly, indicating that residents gradually strengthened their attention to physical health. The coefficients of health in southern China are smaller than those in northern China before and during COVID-19.

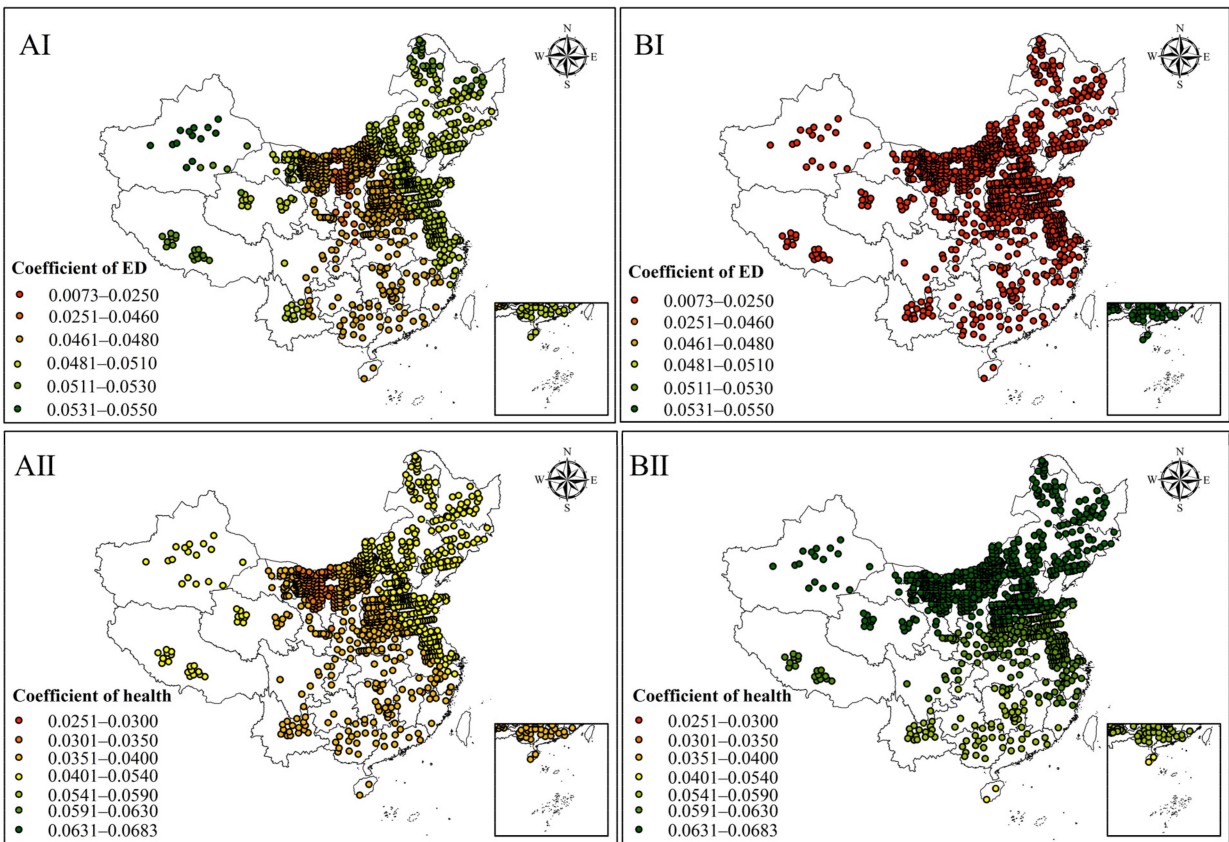

**Figure 7.** The effects of environmental quality (**above**) and environmental effects on health (**below**) in describing WTPEP before (**left**) and during (**right**) COVID-19 based on MGWR models.

4.4.2. Scale Effect Analysis of WTPEP's Drivers

Different driving factors' bandwidths before and during COVID-19 in MGWR models are shown in Table 7. According to the proportion of bandwidth scale in the global sample and corresponding administrative area size, which reflected that driving factors present scale effects at two levels: global scale (BW > 50%) and local scale (BW ≤ 50%). Before COVID-19, the action scale from global to local was ED > age > health > EQ > Edu > gender > (income)$^2$ > income. The bandwidth scales of ED, age, health, EQ, and Edu were all above 504, and the values of age and Edu were above 90%, which can be regarded as macro driving factors at the global level. The bandwidths of gender, (income)$^2$ and income were 261, 65, and 43, respectively, and the BW value was less than 50%, which can be regarded as micro-driving factors at the local level. During COVID-19, the factors significantly impacted on WTPEP also have certain spatial heterogeneity, and the action

scale from global to local was ED > health > EQ > Edu > age > gender > (income)$^2$ > income. Among them, ED, health, EQ, and Edu were macro-drivers, whereas other factors showed local micro effects. The relationships between these factors and WTPEP at the individual level exhibited high spatial non-stationarity.

**Table 7.** Summary statistics for MGWR parameter estimates and bandwidth.

| Variable | Mean | | STD | | Min | | Median | | Max | | MGWR Bandwidth | |
|---|---|---|---|---|---|---|---|---|---|---|---|---|
| | Before | During | Before | During | Before | During | Before | During | Before | During | Before | During |
| Intercept | −0.032 | 0.045 | 0.162 | 0.220 | −0.408 | −0.483 | −0.065 | 0.013 | 0.356 | 0.744 | 86 | 46 |
| income | 0.361 | 0.298 | 0.366 | 0.402 | −0.370 | −0.402 | 0.300 | 0.190 | 1.343 | 1.942 | 43 | 43 |
| (income)$^2$ | 0.055 | 0.211 | 0.490 | 0.564 | −0.890 | −0.403 | 0.113 | 0.160 | 1.191 | 1.979 | 65 | 134 |
| age | 0.008 | −0.063 | 0.003 | 0.074 | 0.005 | −0.279 | 0.007 | −0.055 | 0.020 | 0.194 | 1008 | 195 |
| gender | 0.040 | 0.026 | 0.068 | 0.076 | −0.101 | −0.115 | 0.036 | 0.008 | 0.271 | 0.243 | 261 | 181 |
| Edu | 0.061 | 0.036 | 0.039 | 0.003 | −0.025 | 0.025 | 0.079 | 0.037 | 0.111 | 0.039 | 586 | 1008 |
| EQ | 0.023 | −0.001 | 0.005 | 0.002 | 0.010 | −0.008 | 0.024 | −0.001 | 0.031 | 0.002 | 965 | 1008 |
| ED | 0.048 | 0.014 | 0.002 | 0.002 | 0.046 | 0.007 | 0.048 | 0.015 | 0.055 | 0.018 | 1008 | 1008 |
| health | 0.040 | 0.064 | 0.003 | 0.003 | 0.034 | 0.054 | 0.040 | 0.064 | 0.049 | 0.068 | 997 | 1008 |

## 5. Discussion

### 5.1. Features Comparison of WTPEP before and during COVID-19

WTPEP represents the maximum amount of money that residents are willing to invest in environmental protection. Identifying changes in residents' WTPEP under the epidemic has multiple roles in improving the environment, restoring the ecology, improving well-being, and improving the state of the epidemic. Compared to before COVID-19, WTPEP shows an upward trend during COVID-19, and residents' environmental awareness has generally improved. Further analysis can confirm that this trend has high value in terms of residents' recognition of the environmental damage already done to the planet and the future use of natural resources, but not so strong in specific environmental behaviors, such as recycling. Therefore, even if the epidemic makes individuals more aware of environmental issues, it is not enough to offset the negative externalities generated in this case [58,59]. But this finding of an upward trend shows the initial transition to practical action that can be considered a starting point for environmentally friendly behavior. In the spatial dimension, WTPEP is interlaced and unevenly distributed, which results from the interaction between residents' internal psychological cognition and the pressure brought by the change in the external environment [60]. Residents' internal cognition is the basis of their understanding of the external world, which is shaped by a variety of demographic characteristics of themselves; for example, education has an educational effect on people's environmental awareness [22]. The external environment affects individuals' judgment of objective things, for example, the degree of environmental degradation is positively correlated with WTPEP [16]. While some studies have concluded that the pandemic is more severe than that observed by residents at their locations [61], other studies have demonstrated that people's perceptions of environmental and health issues depend on what they observe in their surroundings [62]. The analysis in this study is consistent with the latter approach, and argues that individuals' assessments of the environmental costs and health costs vary based on their experiences in their locations, more specifically, on changes in local environmental quality and the extent of the COVID-19 pandemic. Therefore, improving residents' WTPEP needs to shape their internal cognition and optimize the external environment. The results are important for investigating residents' pro-environmental behavioral intentions to support environmental protection.

### 5.2. Spatial Heterogeneity Analysis of EKCs

There is spatial heterogeneity in EKCs before and during COVID-19 with different shapes and inflection points of EKCs in different regions, which is mainly due to the trade-off between the availability of environmental public goods and any personal consumption ability. Before COVID-19, the western region had poor resource and environmental en-

dowments, and the central government implemented an ecological compensation policy to enhance residents' willingness to protect the environment, so residents' income levels are on the left side of the inflection points of the inverted U-shaped EKCs (the shape is like the III of Figure 2) [63]. However, as residents' income levels increased, their marginal environmental protection willingness showed a downward trend. Residents' marginal environmental protection willingness would be lower than zero when the increased income was greater than the government's subsidy for residents' environmental protection behavior, and residents' WTPEP would show a downward trend [20]. However, during COVID-19, the government's epidemic prevention measures have had a huge negative impact on social production activities, and residents have a greater pessimistic perception of future income, leading to a reduction in personal consumption capacity [64]. Therefore, residents' WTPEP is negatively correlated with income, and income is located on the left sides of the U-shaped EKCs (the shape is like the I of Figure 2). However, the western region is dominated by the primary and secondary industries, and the ecological environment is closely related to the economic recovery during the epidemic period. Residents have a deeper understanding of environmental protection, so their WTPEP is on the rise when the income increases to the EKC's inflection point value. Therefore, the government of the western region should increase residents' income to improve their ability to improve the environment. However, during COVID-19, residents' WTPEP in the Central Plains showed an inverted U-shaped EKC (the shape is like part III of Figure 2), and incomes were on the left sides of the inflection points, mainly due to differences in economic development levels between the eastern and western regions. Chinese current economic development is largely at the expense of the environment. Thus, residents in economically developed high-pollution areas increase the demand for a clean environment due to the improvement of environmental indicators during COVID-19 [65]. They also have the ability to bear the cost of pollution control. Therefore, with the increase in income, residents' WTPEP is on the rise. Due to the increase in income, residents are more able to protect themselves from environmental pollution, and marginal WTPEP shows a downward trend. When income rises sharply, residents have the ability to avoid the adverse effects of serious environmental pollution by choosing a better working and living environment when income rises sharply, so the marginal WTPEP will fall below zero when income reaches the inflection point value [66]. Therefore, governments in the central plains of region should improve the environmental awareness of high-income people.

The empirical results show that only two developed cities, Beijing and Tianjin, show a positive correlation between residents' income and WTPEP before and during COVID-19 (the shape is like part IV of Figure 2). Balancing the relationship between income and WTPEP is necessary for improving residents' WTPWP. Since the relationships between income and WTPEP showed spatial heterogeneity, it is import to explore an effective path to achieve harmony between residents' income and WTPEP, such as the law of Chinese environmental tax [67]. Combining the empirical results, the future government can further optimize the environmental tax in terms of diversification of levy types and enforceability of levy standards. Residents' environmental awareness determines the extent to which individuals convert income into WTPEP; therefore, region-specific regulations and policies also should be formulated to improve residents' environmental awareness [68].

### 5.3. Key Drivers and Scale Effects of WTPEP

In addition to income, residents' WTPEP is influenced by demographic characteristics, environmental conditions, and health. Up to now, most research hasn't taken a full account of the spatial relationship between WTPEP and drivers. Studies related to WTPEP can be summarized in three points: WTPEP is assessed by social investigation method or modeled in economics, and then the effects of driving factors are analyzed by non-spatial models [69]; the effects of the driving factors are analyzed by spatial autocorrelation models [70]; WTPEP is assessed by social investigation method, and then the effects of the driving factors are analyzed by local spatial regression models [71]. The first type of

research ignores the spatial correlation of drivers and dependent variables. The second type of research considers the spatial correlation of driving factors and dependent variables, but the effect of spatial heterogeneity is ignored [72]. Therefore, the results of these two types of studies are controversial. The third category considers the spatial heterogeneity of influencing factors on WTPEP, but spatial autocorrelation and spatial heterogeneity were not compared and analyzed. To make up for this deficiency, we comparatively analyzed the spatial autocorrelation and spatial heterogeneity of the WTPEP drivers and further studied the scale effects.

The results show that the influence of different factors on WTPEP varies greatly, among which education, environmental degradation, and health are more significant. Education level and WTPEP are positively correlated. Higher education levels significantly increases people's income, and higher education experience may also increase people's attention to environmental protection, so residents may have a stronger WTPEP to reduce negative environmental externalities [73]. Regional education levels significantly affect local residents' WTPEP, so the education gap between western and eastern China becomes a factor in the spatial differences in WTPEP [74]. To eliminate spatial differences in WTPEP, the government needs to optimize education policies, such as optimizing the education budget allocation mechanism, compensating for the benefits, and providing classified guidance for the development of education in less developed and rural areas.

The degree of environmental degradation is positively correlated with WTPEP and affects WTPEP more significantly during COVID-19. Environmental degradation has a serious impact on residents' well-being and material production, and these sunk costs make residents a positive awareness of environmental protection [44]. COVID-19 lockdown improved environmental indicators, whose positive externalities have increased residents' demand for a better ecological environment, so the impact of environmental degradation on WTPEP has increased during COVID-19 [75]. However, this result needs to be examined in the sense that WTPEP enhancement cannot be at the expense of increased environmental degradation, as the goal of WTPEP enhancement is to protect the environment. However, the rise in residents' demand for a better environment during COVID-19 is an opportunity that can be seized. As "participants" and "feedbackers" in the construction of ecological civilization, residents' internal value judgments on the demand for a better environment will, to a certain extent, have an important impact on their environmental behavioral decisions [76]. Therefore, the government should advocate the establishment of a comprehensive concept of ecological well-being and encourage "pro-environmental" behaviors conducive to the construction of ecological civilization.

The impact of environmental degradation on health is positively correlated with WTPEP, and it significantly affects WTPEP during COVID-19. The health problems caused by environmental degradation increase the potential cost of residents' survival (e. g., health cost and monetary cost). In order to maximize benefits, rational people choose to reduce environmental consumption when weighing costs and benefits [77]. During COVID-19, residents' attention to health has increased, resulting in the effect of health on WTPEP being more significant [78]. In developing sustainable action programs, governments usually treat environmental protection differently from health security issues [79]. However, in the context of global environmental change, the gap between environmental concern and health concern will result that the positive externalities of environmental protection on health and the demand for a high-quality environment by residents suffering from diseases will be overlooked. Therefore, the optimization of such positive externalities depends on reducing environmental pollution, improving public health services, and establishing synergistic protection mechanisms.

China provides an extremely attractive venue for investigating the driving factors and scale effects of residents' WTPEP. The results showed that influenced by a combination of drivers and scale effects, major public health events improved residents' WTPEP with a characteristic of spatial heterogeneity. Environmental degradation, health, environmental quality, and education are macro-drivers, and their spatial effect on WTPEP are relatively

stable and wide. Therefore, governments should formulate relevant management strategies to deal with macro-drivers. Local micro effects of income, age, and gender show significant spatial differentiation in their action intensity; hence, governments should develop appropriate regional strategies to deal with micro-drivers. Due to the spatial heterogeneity and scale difference of the influence of each explanatory variable on WTPEP, it is essential to select the appropriate method to deal with multi-scale matching. The overall partition of large watershed scale according to provinces can guide the local policy to narrow the regional gap [80]. In other words, considering the overall situation on a larger scale and implementing more accurate management measures on a smaller scale. The research presented in this paper provides references for other countries facing similar situations to China.

*5.4. Policy Implications*

From this study, we can summarize three policy references.

Raise residents' awareness of environmental protection and activate the power source of environmental protection. The results show that residents' WTPEP is greatly affected by income, education, environment, and health factors. The government should improve macro-control management of environmental protection construction, and strengthen environmental publicity and education to improve residents' environmental awareness level from cognition. Renew residents' consumption concept and strengthen the awareness of ecological consumption and green consumption. Strengthen the environmental ethics education and social responsibility consciousness of high-income groups and reward exemplary environmental behavior.

Innovate the environmental tax collection model, rationally divide the environmental tax collection groups and standards, and enhance the willingness and participation of residents to pay environmental taxes. The rise of WTPEP provides a material basis for the collection of environmental taxes, therefore, the government should improve the fiscal and taxation policy system of environmental tax to realize the scientific collection and accurate use of environmental tax revenues. The application of the environmental tax has improved the efficiency of environmental protection and the quality of the environment, so residents' demand for a better environment has gradually increased, further forming a spiraling situation of harmony between man and nature.

Focus on regional heterogeneity, adjust measures to local conditions, and implement differentiated and applicable environmental policies. The results show that the driving factors of WTPEP have spatial heterogeneity. When formulating environmental policies, the government should fully consider regional characteristics, and avoid unified static standards in various regions or "one size fits all" environmental policies and regulatory intensity. It should adopt flexible and rolling regulatory standards according to regional characteristics and development reality and comprehensively use multiple regulatory forms.

## 6. Conclusions

To analyze changes in Chinese WTPEP before and during COVID-19, this study evaluated Chinese WTPEP based on the questionnaire method at the individual level. The results show that residents' WTPEP has a significant increase during COVID-19 with spatial differences, which indicates that residents' demand for ecological environment is increasing. To clarify whether the EKC relationship exists in WTPEP, this study modeled income and WTPEP basing the theory of EKC, and found that the shapes and inflection points of the relationships between income and WTPEP are spatially heterogeneous before and during COVID-19, which made up for deficiency that current related research was mainly based on non-spatially heterogeneous regression models. The northern WTPEP is larger than the southern, which indicates that there is a spatial imbalance in WTPEP. To explore driving factors and scale effects of WTPEP, this study comparatively analyzed the global spatial regression model (SLM and SEM) and the local spatial regression model (GWR and MGWR) based on the theory of EKC. The results show that environmental

degradation, health, environmental quality, and education are macro-drivers, whereas income, age, and gender show local micro effects. Those factors can help policymakers better understand which factors are more suitable for macro or micro environmental policy-making and what targeted measures could be taken to solve the contradiction between the growing ecological environmental demand of residents and the spatial imbalance of WTPEP in future. Based on the results and discussion, we provide policy references for improving residents' WTPEP and formulating a scientific environmental protection system: raise residents' awareness of environmental protection; innovate the environmental tax collection model; adjust measures to local conditions, and implement differentiated and applicable environmental policies.

However, WTPEP only refers to a potential awareness of environmental protection among residents and does not measure actual environmental behavior. Therefore, our single focus on WTPEP limits the boundaries of the study, and future research could go beyond this limitation to explore the driving mechanisms that translate residents' environmental intentions into actual actions. The perspective of this study is in terms of individual micro-decision-making; however, efficient implementation of environmental policies requires the joint efforts of macro-level government decisions and micro-level will of residents. Future research could explore the driving mechanism of government macro-policy based on an integrated model of government macro-decisions and residents' micro-will. The research of this paper can provide reference for other countries that have the same situation as China.

**Supplementary Materials:** The following supporting information can be downloaded at: https://www.mdpi.com/article/10.3390/ijgi12040163/s1, Supplementary Section S1 "Distribution map of the ratios of sample size to population size in each province"; Supplementary Section S2 "Map of population density distribution in each province"; Supplementary Section S3 "Official data sources".

**Author Contributions:** Conceptualization, Hongkun Zhao and Yaofeng Yang; methodology, Hongkun Zhao; software, Hongkun Zhao; validation, Hongkun Zhao; formal analysis, Hongkun Zhao and Yajuan Chen; investigation, Hongkun Zhao and Yajuan Chen; resources, Yajuan Chen; data curation, Hongkun Zhao; writing—original draft preparation, Hongkun Zhao; writing—review and editing, Hongkun Zhao, Yaofeng Yang, Huyang Yu, Zhuo Chen and Zhenwei Yang; visualization, Hongkun Zhao; supervision, Yajuan Chen; project administration, Yajuan Chen; funding acquisition, Yajuan Chen All authors have read and agreed to the published version of the manuscript.

**Funding:** This research was funded by the National Natural Science Foundation of China (No. 32060317), Inner Mongolia Social Science Foundation (No. 21HQ07), The Fundamental Research Funds for the Inner Mongolia Normal University (No. 2022JBBJ006) and The Initial Foundation of Scientific Research for the introduction of talents of Inner Mongolia Normal University (No. 2018YJRC035).

**Data Availability Statement:** The datasets used and/or analyzed during the current study are available from the corresponding author on reasonable request.

**Acknowledgments:** The authors are grateful to the experts who participated in the survey and the residents for their cooperation and patience in the questionnaire survey. The authors thank the anonymous reviewers for their help in improving this paper.

**Conflicts of Interest:** The authors declare no conflict of interest.

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
