# Peer review of "Driving Factors and Scale Effects of Residents’ Willingness to Pay for Environmental Protection under the Impact of COVID-19"

_ijgi, doi:10.3390/ijgi12040163_

Round 1
Reviewer 1 Report
This manuscript analyzes the change in the Chinese residents' willingness to pay for environmental protection before and during the Covid-19 pandemic. Various statistical methods were applied based on an individual-level questionnaire. The study is original and may be of interest to decision makers in addressing major health and environmental threats. The manuscript is well written and organized, but I have the following comments:
1. It is strongly recommended that the questionnaire results be compared with some real data.
2. The figures and tables need more details and explanations, especially Fig. 2.
3. The order of the tables needs to be corrected.
4. In the conclusion, it is recommended to address the limitations of the study and highlight future recommendations for decision makers in addressing major health and environmental threats.
5. The bibliography is appropriate to some degree, but should be revised to include the missing information.
6. The entire manuscript contains some stylistic errors that need to be corrected.
Overall Recommendation: Reconsider after major revision.
Reviewer 2 Report
The paper should be improved. All results are obtained using the questionary results. Therefore, it is essential to do a relevant search with official or public datasets to provide comparative graphics or tables to assessment the quality of the information provided by the 1000 questionary results.
1. What is the main question addressed by the research? This article describes a research project focused on the design, application, collection, and analysis of data obtained through questionnaires applied over three periods during the critical phases of the pandemic caused by COVID-19. These questionnaires were applied in person or using electronic means within regions of China.
2. Do you consider the topic original or relevant in the field? Does it address a specific gap in the field? The work is interesting because it allows us to observe through a numerical analysis the results caused by the COVID-19 pandemic, especially the money resources allocated to health care (environmental protection).
3. What does it add to the subject area compared with other published material? To accurately answer this question, it is necessary to observe comparisons of the work carried out by the authors with other reports, such as official reports, health institutions, and other research work. Research, specifically with COVID-19
4. What specific improvements should the authors consider regarding the methodology? What further controls should be considered? It is essential to study in-depth and define specific criteria related to the questionnaires applied. For example, what was the procedure for choosing the regions where the questionnaire was applied? The difficulty in applying this type of questionnaire is understandable, considering the restrictions (quarantines) that took effect in recent years. However, it is essential to appreciate a statistical procedure for applying the questionnaires (distribution and population density in the regions). Furthermore, what is the level of reliability of the answers obtained? An example could be the average level of studies obtained when applying the questionnaire and comparing it with official reports.
5. Are the conclusions consistent with the evidence and arguments presented and do they address the main question posed? Partially, it is necessary to reinforce the conclusions with the points described in the previous question.
6. Are the references appropriate? Yes, they are adequate.
7. Please include any additional comments on the tables and figures.
In figure 1, it should be considered that the coordinates are obtained with two parameters: latitude and longitude. The size used in the subfigures of Figure 4 does not allow to appreciate the details of each section to be appreciated. It is recommended to increase the resolution or use a version in vector format. The same applies to Figures 5, 6 and 7.
Reviewer 3 Report
-abstract: highlight the significant value of the WTPEP based on the empirical/quantitative value from findings. the verdict given in line 25 - 26 sounds general. Which factor should be emphasized for future research?
- regional overview: please elaborate how the current studies inspired by figure 1 (eg: zoning based on the climate condition? or the case numbers of Covid-19? etc). How the regional overview determine the sample size of this study?
line 142 - 154 please includes the location/region based on the given figure 1 description. How author determine the sample? Does the 1,019 respondents distributed evenly around the region? how to determine the significant of the dataset? Please justify according to the ratio of the numbers of respondents and studies coverage area.
- Theoretical model setting: describe and elaborate further more th EKC model in general. How does the inverted U-shaped relationship between economic development and environmental degradation affecting WTPEP? Is there any indirect relationship between the model and WTPEP?
- Discussion is well described. However, it could be better if the analysed results could be used to explain the significance of the studies.
- Conclusion: explain the future recommendation based on the limitation of the current studies.
Reviewer 4 Report
The research presents insightful information regarding the connections between significant public health incidents, environmental preservation, and people's well-being. Ordinary least squares, the spatial lag model, the spatial error model, the geographically weighted regression, and the multiscale geographically weighted regression are just a few of the models that the authors used in their thorough analysis. The study's findings are presented plainly and lend credence to the authors' main thesis.
It would be beneficial to discuss the willingness to pay endogeneity (WTPEP) in relation to the discussed driving factors in order to improve the article. This would give a more accurate and nuanced understanding of how these variables relate to one another. To ascertain whether the findings can be extrapolated to other populations or regions, it would also be crucial to discuss external validity.
Although there are a few instances of phrasing and grammar errors that could be corrected, the writing is generally clear and concise. Nevertheless, the writing offers a thorough and instructive overview of the idea of WTPEP and the driving forces that affect it as a whole, and the use of examples and references lends credibility to the discussion.
The introduction section provides a clear overview of the research topic but would benefit from a more detailed explanation of the theoretical framework. The results section presents the findings in a clear and logical manner, but a more detailed discussion on the implications of the results would be helpful. The conclusion section summarizes the main findings and provides insights for policymakers but could benefit from more detailed discussion on the limitations of the study and suggestions for future research.
Round 2
Reviewer 1 Report
I believe that the manuscript has been sufficiently improved. I recommend the manuscript for publication in IJGI.
Reviewer 2 Report
All required changes have been made. Therefore, the paper is ready for publication.
Reviewer 4 Report
I believe the manuscript has been largely improved. The authors have done a great work on revision.